**Data Availability Statement:** Precipitation and land surface temperature data are publicly available and can respectively be found at https://gpm.nasa.gov/missions/GPM and https://modis.gsfc.nasa.gov/

# Climate-driven models of leptospirosis dynamics in tropical islands from three oceanic basins

Léa Douchet[1,2]*, Christophe Menkes[1], Vincent Herbreteau[2,3], Joséphine Larrieu[1], Margot Bador[4], Cyrille Goarant[5,6�], Morgan Mangeas[1�]

1 ENTROPIE, IRD, Univ Reunion, CNRS, IFREMER, Univ Nouvelle Calédonie, Nouméa, New Caledonia, 2 ESPACE-DEV, IRD, Univ Montpellier, Univ. Antilles, Univ Guyane, Univ Réunion, Phnom Penh, Cambodia, 3 Institut Pasteur du Cambodge, Epidemiology and Public Health Unit, Phnom Penh, Cambodia, 4 CECI Université de Toulouse, CERFACS/CNRS, Toulouse, France, 5 Institut Pasteur in New Caledonia, Leptospirosis Research and Expertise Unit, Nouméa, New Caledonia, 6 Public Health Division, The Pacific Community, Nouméa, New Caledonia

‑ These authors contributed equally to this work.
* lea.douchet@ird.fr

## Abstract

### Background

Leptospirosis is a neglected zoonosis which remains poorly known despite its epidemic potential, especially in tropical islands where outdoor lifestyle, vulnerability to invasive reservoir species and hot and rainy climate constitute higher risks for infections. Burden remains poorly documented while outbreaks can easily overflow health systems of these isolated and poorly populated areas. Identification of generic patterns driving leptospirosis dynamics across tropical islands would help understand its epidemiology for better preparedness of communities. In this study, we aim to model leptospirosis seasonality and outbreaks in tropical islands based on precipitation and temperature indicators.

### Methodology/Principal findings

We adjusted machine learning models on leptospirosis surveillance data from seven tropical islands (Guadeloupe, Reunion Island, Fiji, Futuna, New Caledonia, and Tahiti) to investigate 1) the effect of climate on the disease's seasonal dynamic, *i.e.*, the centered seasonal profile and 2) inter-annual anomalies, *i.e.*, the incidence deviations from the seasonal profile. The model was then used to estimate seasonal dynamics of leptospirosis in Vanuatu and Puerto Rico where disease incidence data were not available. A robust model, validated across different islands with leave-island-out cross-validation and based on current and 2-month lagged precipitation and current and 1-month lagged temperature, can be constructed to estimate the seasonal dynamic of leptospirosis. In opposition, climate determinants and their importance in estimating inter-annual anomalies highly differed across islands.

### Conclusions/Significance

Climate appears as a strong determinant of leptospirosis seasonality in tropical islands regardless of the diversity of the considered environments and the different lifestyles across

data/dataprod/mod11.php. The leptospirosis raw data were provided by several countries who have not provided permission for it to be publicly shared. For New-Caledonia, the leptospirosis data can be requested by writing to the department of Health and Social affairs of New Caledonia (dass@gouv.nc). For Guadeloupe, a demand should be addressed to the Regional Office of the French Institute for Public Health surveillance Antilles-Guyane (antilles@santepubliquefrance.fr). Data from Reunion Island and Mayotte can be requested through the the French National Reference Center for leptospirosis (spiroc@pasteur.fr). Data for Tahiti can be requested to the Direction of Health of French Polynesia,Health surveillance Office (veille. sanitaire@administration.gov.pf). For Futuna, data may be asked to the Wallis & Futuna health authorities upon request to Deputy Director Public Health, Agence de Santé et l'Hopital de Sia, B.P. 4G Mata'Utu, 98600 UVEA, Wallis (da.sp@adswf.fr). Leptospirosis data for Fiji can be requested to the Ministry of Health by filling a data request form (https://www.health.gov.fj/wp-content/uploads/ 2014/05/Data-Request-Form.pdf) and sending it to the contact provided on the form.

**Funding:** This study was funded by Pacific Fund (Fonds de coopération économique, sociale et culturelle pour le Pacifique, french government)(M. M., C.M.) and ECOMORE 2 (funded by the Agence Française de Développement and coordinated by Institut Pasteur)(V.H., M.M., C.M.). This project has received funding from the European Union's Horizon 2020 research and innovation program under the Marie Sklodowska-Curie Grant agreement No101027577 (M.B.). The funders had no role in study design, data collection and analysis, decision to publish, or preparation of the manuscript.

**Competing interests:** The authors have declared that no competing interests exist.

the islands. However, predictive and expandable abilities from climate indicators weaken when estimating inter-annual outbreaks and emphasize the importance of these local characteristics in the occurrence of outbreaks.

## Author summary

Tropical islands are particularly vulnerable to leptospirosis outbreaks. Hot and rainy climate, abundance of reservoir species and outdoor lifestyle contribute to the high risk for human infection. These isolated areas also deal with difficulties associated with diagnosis because of low awareness of the medical staff, non-specific symptoms of leptospirosis and limited availability of laboratory testing. Leptospirosis remains poorly documented, and a better understanding of its dynamics and its climate drivers would help improve awareness and preparedness of the public health services. In this study, we provide a climate-based model of leptospirosis seasonal dynamics in 7 tropical islands. The use of climate variables from publicly available satellite data makes the model expandable to predict leptospirosis seasonal dynamics in other tropical islands where the disease is not routinely monitored. This study emphasizes the importance of rainfall and temperature in driving the seasonality of leptospirosis in tropical islands. However, climate alone did not appear to not be a sufficient indicator to predict interannual variations, suggesting that the risk of leptospirosis outbreaks must be refined, considering local specificities as the lifestyle and the very local environment.

## Introduction

Leptospirosis, a bacterial infection caused by a spirochete of the genus *Leptospira*, is one of the most widespread zoonotic diseases causing over 1 million cases yearly [1]. Most cases are reported in tropical regions with about 70% of the annual cases under these latitudes [1]. Despite a burden comparable to schistosomiasis, leishmaniasis or lymphatic filariasis, leptospirosis remains neglected and receives insufficient research attention [2].

Asymptomatic carriers, commonly rodents and livestock, shed pathogenic leptospires through their urine after they multiply in kidney tubules for their entire lifespan [3]. Once released into water and soil, leptospires can survive from one week to months in the environment [4]. Although humans can get infected through direct contact with a reservoir animal, most contaminations occur indirectly, after contact with a previously contaminated environment. The bacteria get entry into the body through mucous membranes as nose, mouth, and eyes and through cuts and abrasions of the skin. The time between exposure and the onset of symptoms ranges from 2 days to 4 weeks [3]. Although leptospirosis is commonly benign with few or no clinical manifestations, patients may develop a flu-like illness difficult to diagnose clinically that can drift to severe and potentially fatal Weil's disease or the severe pulmonary hemorrhage [3]. The mean case fatality rate was estimated to be 6.85% globally, but can reach 30% in some developing countries [1].

While modelling the global burden of leptospirosis, Costa *et al.* identified tropical islands as a particular high-risk setting for human infections [1]. Tropical islands are indeed more vulnerable to invasive species like rodents as they are delicate ecosystems with a generally lower biodiversity but higher levels of endemism [5]. Living outdoors also increases exposure to leptospirosis in the islands.

As a water-borne disease, leptospirosis is affected by climate conditions. Tropical hot and humid environments favor the survival of pathogenic leptospires [3], while heavy rainfall and flooding lead to a greater exposure to contaminated water [4,6] and increased contacts between rodents and human populations. Although the effect of floods is not well characterized and depends on the study setting, the considered territory and the scale of the study, a review encompassing 14 case-control and cohort studies identified floodings as a significant risk factor for leptospirosis with an overall odds ratio of 2.19 [7]. Many studies evidenced rainfall as the main climate contributor to leptospirosis outbreaks worldwide [6], including in tropical islands [8–12]. However, no consensus was reached on the time lag between a rain event and an increase in the number of cases reported by the health system, or on the direction of the association (positive or negative). For example, a long-term association with lags of 8 to 10 months was demonstrated in Thailand [13], while rainfall is expected to increase incidence with a lag of one week in Colombia [14]. In the wet zone of Sri Lanka, incidence was positively associated with rainfall with a lag of 5 months, but negatively associated with this same indicator with a lag of 2 and 3 months [15]. In tropical islands, a strong relationship between leptospirosis incidence and rainfall with a two-month time lag was evidenced in Reunion Island [10], French Polynesia [8] and in Futuna [11]. Studies in Mayotte supported a 3-month lag between cases and rainfall and the intensity of outbreaks was associated with the number of consecutive rainy months rather than rainfall amount [9]. In Fiji, the maximum rainfall in the wettest month contributed to model the spatial distribution of leptospirosis seroprevalence [12]. Temperature is also considered an important climatic factor for leptospirosis [3].A warm environment can increase the risk of exposure by attracting animals and humans to the same water sources and by encouraging water-based activities [6,16]. In Reunion Island, temperature of the current month appeared as a great explanatory variable of leptospirosis [10]. Inter-annual increases in incidence in New Caledonia were linked with La Niña phases of the ENSO (El Niño Southern Oscillation), characterized locally by a local hotter and wetter climate [17]. Similarly, in Guadeloupe, a four-fold increase in incidence between 2002 and 2004 occurred simultaneously with two El Niño events [18] that brought unusual weather conditions.

The population of the islands are more at risk, as they live in a tropical climate and generally have difficult access to healthcare (e.g., due to hilly topography and poverty) [6]. Therefore, outbreaks can rapidly overflow health systems. In addition, leptospirosis is poorly notified and poorly documented in many tropical countries, and in tropical islands in particular, mainly because of a lack of medical awareness and difficulties associated in diagnosis as the unavailability of laboratory testing and the non-specific symptoms confound with other tropical infections [19]. Better understanding of leptospirosis' dynamics and environmental triggers would promote the preparedness and awareness of communities and public health services. Studies of leptospirosis in tropical islands, conducted independently, generally differ in terms of available data and methods of investigations, making them difficult to compare and preventing any extension to other territories for which leptospirosis is poorly documented [10,20]. A generic model, *i.e.*, a model validated across different islands, is therefore needed not only to help identify common features that favor infections, but also to highlight differences that can lead to unexpected local outbreaks. Such a model also could be used to extrapolate predictions and therefore inform on the disease burden in areas where it is not monitored.

In this study, we investigated the relationship between climate and leptospirosis incidence in seven tropical islands from three oceanic basins: namely the South-Pacific (New Caledonia, Fiji, Tahiti and Futuna), the Indian Ocean (Reunion and Mayotte) and the North Atlantic (Guadeloupe in the West Indies). These islands are of various scales and are differently affected by leptospirosis. However, they all have tropical to subtropical climates with heavy rainfall during their respective warm seasons. We used satellite data for precipitation and temperature to

study the relationship between climate and leptospirosis in tropical island settings. The developed model assisted in providing estimates of leptospirosis seasonal profile in Vanuatu (South-Pacific Ocean) and Puerto Rico (Caribbean Sea), two tropical islands where the disease is recognized as a health threat, but its dynamic remains poorly documented [21,22].

## Methods

### Islands characteristics

The study area encompasses seven tropical/sub-tropical islands for which leptospirosis surveillance records were available. Four are located in the South-Pacific Ocean (Fiji, New Caledonia, Tahiti and Futuna), two in the Indian Ocean (Mayotte and Reunion Island) and one in the northern hemisphere, in the Caribbean Sea (Guadeloupe) (Fig 1). We added two other tropical islands with comparable climate conditions for which no leptospirosis surveillance records were available: Vanuatu located in the South-Pacific Ocean and Puerto Rico in the Caribbean Sea. All considered islands substantially differ in size with New Caledonia and Fiji being the largest islands (with about 18 000 $km^2$) and Futuna the smallest (with 46 $km^2$). They also have very different populations, from 3,613 inhabitants in Futuna to 3,285,874 in Puerto Rico, a number far above that of Fiji, the second most populated island studied with 855,113 inhabitants (Table 1).

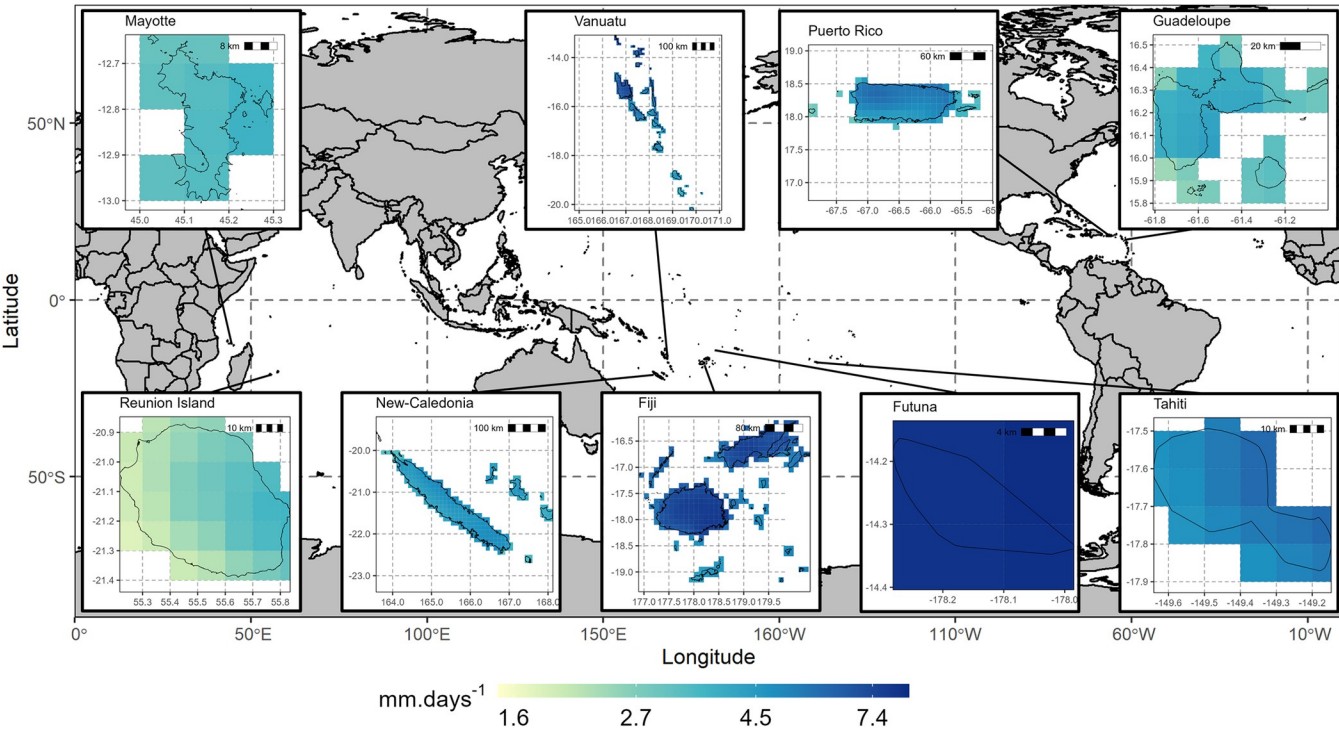

**Fig 1. Mean monthly rainfall rate (mm) from IMERG gridded satellite data in New Caledonia, Fiji, Tahiti, Reunion Island, Mayotte, Futuna, Vanuatu, and Puerto Rico. Rainfall was averaged over the period of leptospirosis data for each island (see Table 1).** Baseline map made with Natural Earth. Free vector and raster map data @ naturalearthdata.com and contains information from OpenStreetMap and OpenStreetMap Foundation, which is made available under the Open Database License (https://opendatacommons.org/licenses/odbl/). No changes where made to the original map. Guadeloupe, Reunion island and Mayotte administrative divisions were downloaded from https://data.humdata.org. New-Caledonia administrative divisions were provided by Kontur (https://www.data.gouv.fr). Tahiti boundaries were provided by the Section Cadastre-Topographie de la Polynésie française (https://www.data.gouv.fr). Futuna boundaries were provided by SPC—SDD Statistics for Development (https://pacificdata.org). Vanuatu administrative boundaries were provided by Secretarial of the Pacific Community (SPC), Statistics for Development Division, Fiji boundaries were provided by Fiji Islands Bureau of Statistics, Pacific Catastrophe Risk Assessment and Financing Initiative (PCRAFI) and Puerto Rico boundaries were provided by GADM project (https://data.humdata.org).

**Table 1. Characteristics of New Caledonia, Fiji, Tahiti, Reunion Island, Mayotte, Futuna, Guadeloupe, Vanuatu and Puerto Rico.**

| Island | Monthly precipitation (±S.E.)[a] | Monthly temperature (±S.E.)[a] | Population[b] | Leptospirosis study period | Mean monthly number of cases ±S.E. (median)[c] |
|---|---|---|---|---|---|
| New Caledonia | 135.3±104.1 | 25.1±3.0 | 251,149 | 2000.06–2021.07 | 6.8±9.1 (3) |
| Fiji | 198.9±137.9 | 25.5±1.6 | 855,113 | 2003.01–2017.02 | 18±26 (8) |
| Tahiti | 160.4±137.8 | 24.1±1.3 | 181,672 | 2006.01–2016.12 | 4.9±3.5 (4) |
| Reunion island | 79.7±100.3 | 23.6±2.9 | 812,948 | 2000.06–2020.12 | 5.8±5.9 (4) |
| Mayotte | 100.2±114.8 | 26.1±1.3 | 211,480 | 2007.01–2020.12 | 8.7±12.5 (3) |
| Futuna | 254.5±193.2 | 24.1±1.3 | 3,923 | 2004.01–2014.12 | 2.8±3.1 (2) |
| Guadeloupe | 103.7±83.2 | 26.8±1.4 | 394,526 | 2011.03–2020.12 | 8.8±7.1 (7) |
| Vanuatu | 169.1±107.2 | 24.0±1.4 | 298,333 | X | X |
| Puerto Rico | 146.0±93.6 | 27.7±1.5 | 3,285,874 | X | X |

[a]Monthly mean temperature (˚C) and mean monthly precipitation (mm) with associated standard error (S.E.) were computed for each island.

[b]The population was averaged over the leptospirosis study period except for Vanuatu and Puerto Rico for which population corresponds to the year 2020.

[c]Mean and median number of cases were computed over the study period in New Caledonia, Fiji, Tahiti, Reunion Island, Mayotte, Futuna and Guadeloupe. For all these islands, the minimum number of cases per month was zero.

All islands except Mayotte and Futuna present high orographic contrasts, especially Reunion Island (highest point 3,070 m) and Tahiti (highest point 2,241 m). Futuna is by far the most watered island and has highly variable precipitation throughout the year. In opposition, Reunion Island is the driest island and has the lowest average temperatures (Table 1). Mayotte, Guadeloupe, and Puerto Rico have the warmer weather with mean temperature above 26˚C.

## Leptospirosis data

We collected leptospirosis monthly records of laboratory confirmed cases available for the seven islands: Mayotte, Guadeloupe, Reunion Island, Fiji, New-Caledonia, Tahiti, and Futuna (see acknowledgements). Leptospirosis data for New Caledonia were provided by the Institut Pasteur of New Caledonia, which performed routine diagnostic activities as part of public health surveillance until 2016 [17] and were completed with recent time series from the Government of New Caledonia. The French National Reference Center for leptospirosis provided leptospirosis surveillance records for Mayotte and Guadeloupe. The Direction of Health of French Polynesia provided leptospirosis time series for Tahiti. Leptospirosis data for Fiji, issued from laboratory-based surveillance, were provided by the Fiji Ministry of health. Leptospirosis data for Reunion [10] and for Futuna from Wallis and Futuna Health Agency [11] were available from previously published articles and updated from the French National Reference Centre.

Leptospirosis records spanned from 2000 to 2021 with the longest series for New Caledonia (21 years) and the shortest for Guadeloupe (9 years). Tahiti and Futuna had 10-year series while Mayotte, Fiji and Reunion Island respectively had 13-, 14 and 20-year series (Table 1 and Fig 2).

Leptospirosis time series of monthly number of cases were converted into monthly incidences. We computed the seasonal profile of incidence for each island by taking the median yearly profile, *i.e.*, the median incidence of each month (S1 Fig). Since data were not normally distributed, the use of a median profile allowed us to provide a stable estimate for seasonality that was less sensitive to extreme values and the long end tail of the incidence distribution. We

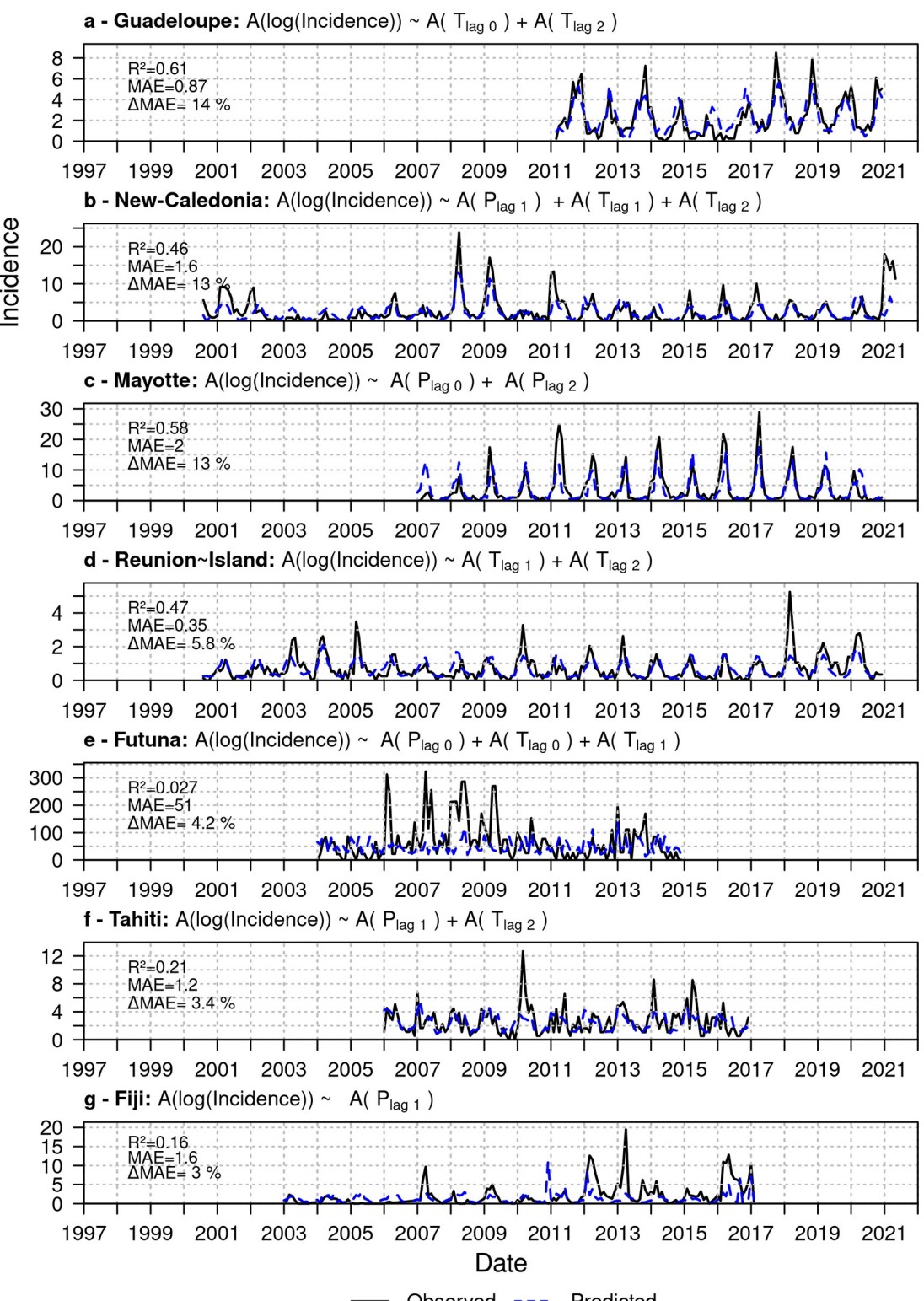

**Fig 2. Observed and predicted leptospirosis dynamics obtained by cross-validation.** For each island, we selected the best predictive model to estimate log-transformed leptospirosis inter-annual variability (A(log(incidence))) based on temperature (T) and precipitation (P) anomalies with a lag of 0 to 2 months. We retrieved the dynamics by summing the modelled anomalies of log-incidence obtained with leave-year-out cross-validation to the seasonal profile of log-incidence and by performing an exponential transformation to draw incidence. The $\Delta MAE$ is the gain in $MAE$ compared to the seasonal profile.

then retrieved this seasonal profile to the time series of incidences to compute the inter-annual anomalies. We repeated these same operations on the log-transformed dataset of leptospirosis monthly incidence to compute the seasonal profile and inter annual anomalies of log-incidence.

## Precipitation and temperature from satellite data

Monthly rainfall data from satellite images were retrieved from the IMERG final product of the Global Precipitation Measurement (GPM) mission (https://gpm.nasa.gov/missions/GPM). This gridded dataset (0.1˚x0.1˚ spatial resolution) provided monthly estimates of rainfall rate ($mm.hr^{-1}$) from 2000 to present (Fig 1). These rates were transformed to cumulative monthly rainfall ($mm$) and averaged spatially over each island to match the dimensions of epidemiological data. Monthly land surface temperatures (˚$C$) were retrieved from the Terra Moderate Resolution Imaging Spectroradiometer (MODIS) version 6 database (https://modis.gsfc.nasa.gov/data/dataprod/mod11.php) from May 2000 to September 2021 at the spatial resolution of 0.05˚ x 0.05˚. The gridded land surface temperature was averaged spatially over each island.

We defined the seasonal profiles of temperature and precipitation of each island classically as the composite year where the climatological month is the average of the same calendar month across the entire time series (S1 Fig). We then subtracted these climatologies to their respective time-series to compute the inter-annual anomalies of precipitation and temperature.

## Statistical analysis and validation

**Support Vector Regression fitting.** We prepared data and conducted statistical analyses with R software (http://www.rproject.org/) using raster and sf packages. We performed a principal component analysis on climate and leptospirosis anomalies to explore the relationship between variables, especially between leptospirosis anomalies and climate variables. Then, we adjusted Support Vector Regression (SVR) on the leptospirosis dynamics with precipitation ($P$) and temperatures ($T$) at lag 0, 1 and/or 2 months ($lag\ x, x \in \{0,1,2\}$)) as explanatory variables. SVR model appears as a great tool to model complex nonlinear relationships and interactions without prior knowledge on the shape of the links between the explanatory and response variable. As a non-parametric supervised machine learning model, it doesn't rely on any assumptions regarding the data distribution as normality, independence, and homoscedasticity of the observations, and remains robust to collinearity issues [23]. We adjusted two different models:

1. the seasonal model encompassed Reunion Island, Guadeloupe, Tahiti, Fiji, New Caledonia and Mayotte and aimed at modelling the normalized (centered) seasonal profile of leptospirosis based on the normalized seasonal profile of temperature and precipitation (respectively $S(P_{lag\ x})$ and $S(T_{lag\ x}), x \in \{0,1,2\}$ months)).

2. the inter-annual model aimed at estimating the anomalies of the log-transformed incidence based on the precipitation and temperature anomalies (respectively $A(P_{lag\ x})$ and $A(T_{lag\ x}), x \in \{0,1,2\}$ months). We adjusted this model on each island independently.

For both models, we explored all combinations of variables to find out the model that best predicts the dynamics of leptospirosis, *i.e.*, provides the best results in Cross-Validation (CV). Then, we further adjusted the hyperparameters of the gaussian kernel, the cost and the gamma parameters, using grid searching and based on Leave-Year-Out (LYO) CV for the inter-annual model and Leave-Island-Out (LIO) CV for the seasonal model.

**Models' analysis and prediction.**   The quality of the model's adjustment, *i.e.*, the difference between the observed incidence *y* and the estimated one $\hat{y}$ for each time *t* of the series *N*, was assessed with 2 criteria: the $R^2$ computed as $R^2 = cor(y,\hat{y})^2$ and the Mean Absolute Error (MAE) computed as $(\Sigma_{t=1}^{N}|y_t - \hat{y}_t|)/N$. We computed an additional quality criteria for the inter-annual model, the $\Delta MAE$ (%) to quantify the improvement of the MAE score from model estimation compared to a prediction solely based on the seasonal profile, this additional quality criteria was computed as *ΔMAE = 100\*(MAEprofile-MAE)/MAEprofile*. Among all tested models, the best seasonal and inter-annual models were respectively selected based on the lowest *MAE* and highest *ΔMAE* score*s* obtained by cross-validation.

We further analyzed the models' outputs by computing the importance score of each of its explanatory variables. We define the importance score as the mean increase in MAE induced by 1000 permutations of the variable's value within the dataset, *i.e.*, $MAE_{permuted}/MAE_{not\ permuted}$. We also studied the marginal effect of each variable in the SVR model. For each island, we computed the Individuals Conditional Expectancy (ICE) by varying one variable into its range of values and drawing the resulting predicted incidence of leptospirosis, the other variables being held constant. We estimated the marginal effect of each variable by averaging the ICEs across islands.

We studied the spatial predictive abilities of the seasonal model by performing LIO CV. We further used the seasonal model to predict the leptospirosis seasonal dynamics in Puerto Rico and Vanuatu islands. Finally, we performed a LYO CV on the inter-annual model to assess the model's ability to predict future positive and negative anomalies.

## Results

### Leptospirosis dynamic in tropical islands

We studied seven tropical islands, Reunion, Guadeloupe, Tahiti, Fiji, New Caledonia, Mayotte, and Futuna for which leptospirosis is routinely monitored by health services. Futuna had by far the highest and the most variable incidence, with an averaged incidence ± standard error of 68.9±72.8. Four consecutive peaks of incidence above 250 cases per 100,000 inhabitants occurred between 2006 and 2010 (Fig 2E). In opposition, Reunion Island held the lowest and less variable incidence (0.71±0.72). This island recorded its highest incidence (5.25) year 2018 (Fig 2D). In Guadeloupe (incidence of 2.3±1.8), we observed the lowest incidence between 2014 and 2016, whereas high peaks of incidence recurrently occurred during the rainy season of the other years (Fig 2A). Although Fiji and Tahiti had a low average incidence (respectively 2±3 and 2.7±1.9), both islands observed outbreaks far above the baseline with respective incidence of 20 (year 2013 in Fiji) and 12.6 (year 2010 in Tahiti) (Fig 2F and 2G). New Caledonia (incidence of 2.72±3.6) and Mayotte (incidence of 3.8±5.4) seasonally reached high incidence and inter-annual outbreaks drastically level up this incidence with six peaks and four peaks above 17 cases per 100,000 inhabitants respectively (Figs 2B, 2C and 3).

In Reunion Island, Tahiti, Fiji, New Caledonia, and Mayotte, we could break down the seasonal dynamics in two distinct parts. The first six to seven months of the year (from January to June/July) were characterized by a high incidence whereas incidence decreased the 6 remaining months (from June/July to December) (Figs 3 and S1). Guadeloupe, located in the northern hemisphere, showed this same pattern but the highest incidence occurred later in the year and for a longer period, from March to November. Mayotte showed the highest seasonal amplitude of incidence with a maximum of 12.9 cases per 100,000 reached in April. This peak belonged to a seasonal burst in incidence spanning a short period of time of about 3 months. In opposition, Reunion Island recorded the lowest seasonal amplitude (maximum incidence of 1.5). In Tahiti, the month with the highest incidence (February with an incidence of 4.5) is

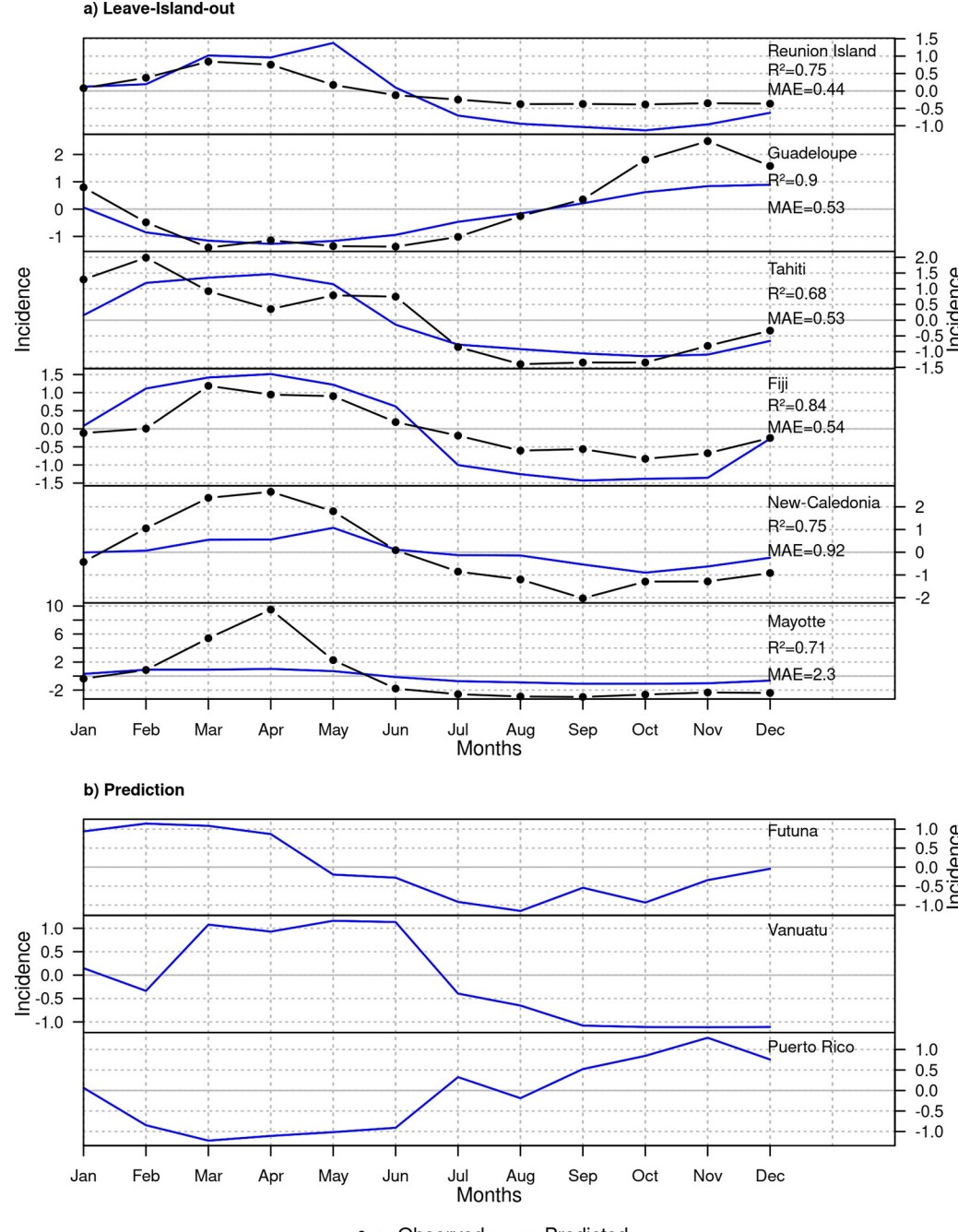

**Fig 3. Observed and predicted normalized seasonal profile of leptospirosis.** We adjusted a global SVR model encompassing Reunion Island, Guadeloupe, Tahiti, Fiji, New Caledonia, and Mayotte with normalized seasonal precipitation at lag 0 and 2 and temperature at lag 0 to 1 months as explanatory variables. The predicted seasonal profile (blue line) was obtained using a) cross-validation on the island used to build the model and b) prediction of the global model for the island not included in the model. The observed seasonal profile (black line with dots) was obtained by computing the median incidence each month.

followed by a less intense incidence rebound (3.3) occurring 3 months later (Figs 3 and S1). Futuna did not show a clear seasonal pattern with incidence varying a lot throughout the year (Figs 2 and S1).

## Leptospirosis seasonality

**Climate determinants of seasonal leptospirosis.** We first focused on the seasonal dynamics of leptospirosis. Since Futuna did not show a clear seasonal pattern, we removed this island from our seasonal model and solely studied its inter-annual anomalies. We tested all combinations of climate variables to identify the model that best predicted the seasonal variation in tropical islands (S2 Fig). This process helped identify four variables driving leptospirosis seasonally: the current-month and 2-months lagged precipitation, and the current and 1-month lagged temperature (Fig 4). Among them, precipitation-based variables had the highest importance in the model, especially with a 2-month lag (importance score of 1.72) (Fig 4A). Wettest months led to higher predicted incidences (Fig 4B). This effect was stronger and homogeneous across the islands (low standard error) when considering the precipitation with a lag of 2 months (Fig 4B) than with a lag of 1-month. The two temperature variables had similar low importance scores of about 1.13 (Fig 4A) and both very high and very low temperatures seemed to increase incidence while values close to average temperature lowered incidence (Fig 4B). This effect was however highly variable across islands (high standard error, Fig 4B).

**Predicted seasonality.** The selected model including precipitation (lag 0 and 2 months) and temperature (lag 0 and 1 month) accurately predicted leptospirosis seasonality with a global MAE in leave-island-out CV of 0.87. For all islands, the seasonal dynamics was well captured by our model with $R^2$ above 0.68 for all islands (Fig 3A). The predicted leptospirosis seasonality in Guadeloupe, Reunion Island, Tahiti and Fiji quantitatively matched the observed data. We could nevertheless observe slight differences between observed and predicted incidence in these islands. Leptospirosis was underestimated during the seasonal peak in incidence in Guadeloupe with a difference of 1.65 between observed and predicted maximum incidence. In opposition, we observed an overestimation of incidence in Reunion Island and Fiji during the high season of leptospirosis that was followed by an underestimation for the rest of the year (Fig 3A). Despite the good adequacy between the model and the observations in Tahiti (with a MAE of 0.53), predicted dynamic did not reproduce the two peaks in incidence occurring in February and May. The model did moreover not capture the sharp dynamic observed

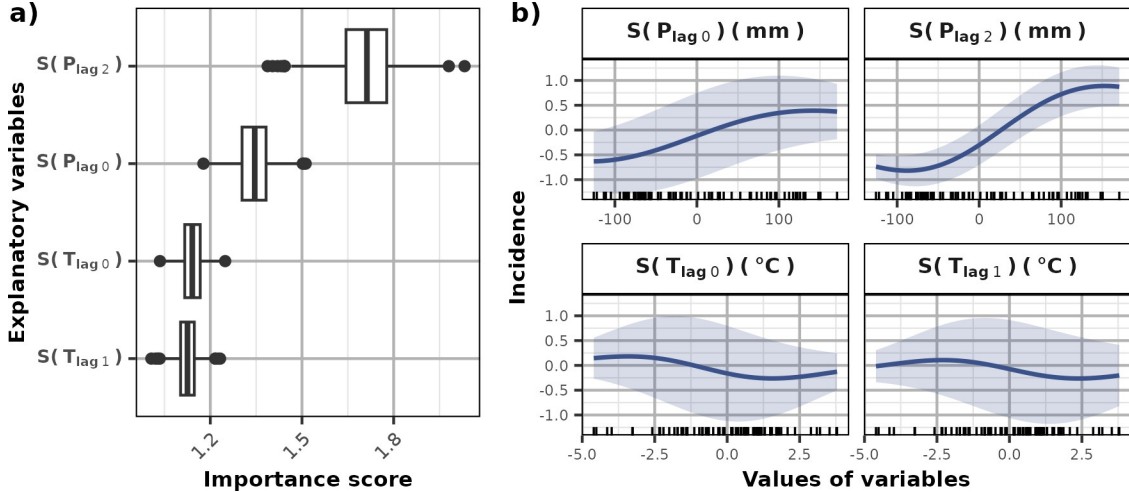

**Fig 4. Effect of the explanatory variable in the SVR seasonal model.** The importance score (a) represents the increase in MAE caused by repeated permutations of the variable in the dataset. The partial dependence curves (b) estimate the marginal effect of each variable by computing the average (blue line) (± standard error, blue ribbon) of Individuals Conditional Expectancy (ICE) curves (see methods).

in Mayotte and provided a much smoother seasonality that strongly underestimated leptospirosis during the seasonal peak in incidence from February to May (MAE of 2.3) (Fig 3A). In New Caledonia, the model shrank the amplitude of leptospirosis seasonality with observed and estimated normalized incidence ranging from −2 to 2.7 and from −0.9 to 1.1 respectively (Fig 3A).

We then used the model to estimate leptospirosis seasonality in Futuna, Vanuatu and Puerto Rico. For these islands, the predicted seasonal dynamic could be split into two periods: a period of high incidence lasting about 6 months and a period of low incidence for the 6 remaining months (Fig 3B). The amplitude of the predicted dynamic was similar for all islands, with an amplitude (maximum incidence—minimum incidence) of 2.3 for Futuna, 2.3 for Vanuatu and of 2.5 for Puerto Rico. In Vanuatu and Futuna, high incidence would occur at the beginning of the year. Leptospirosis incidence in Vanuatu is predicted to reach a peak in May/June. This maximum would be slightly later in the year than in the other islands of the South Pacific Ocean (Tahiti, Fiji and New Caledonia) while it is reached earlier in the year, in February, in Futuna. In opposition, Puerto Rico would have the highest incidence during the last 6 months of the year and reach its maximum in November similarly to Guadeloupe (Fig 3B).

## Inter-annual outbreaks in tropical islands

**Climate determinants of leptospirosis outbreaks.** We studied leptospirosis anomalies compared to the seasonal profile in each island. We performed PCA to obtain a simplified understanding of the link between inter-annual variability of log incidence and climate data. This analysis brought into light different patterns of correlations between variables (Fig 5). Anomalies of leptospirosis incidence appeared to be affected by different sets of variables for each island, compromising the adjustment of a generic model. Considering the diversity of relationships that we observed between leptospirosis anomalies and climate data, we were not able to identify common determinants to build a global model encompassing all islands. We therefore tested all combinations of variables to model leptospirosis inter-annual variability independently for each island (S3 Fig).

The inter-annual variability of leptospirosis incidence were well represented by the first two dimensions of the PCA in Fiji, Guadeloupe, Mayotte, New Caledonia, and Reunion Island. In Reunion Island, Guadeloupe and New Caledonia, temperature negatively correlated with anomalies whatever the lag, while we observed poor or no correlation in Mayotte (Fig 5). No correlation with temperature appeared while analyzing the data globally (Fig 5). Precipitation poorly correlated with incidence anomalies in Reunion Island and the inter-annual leptospirosis dynamic was solely driven by temperatures (lags of 1 and 2 months) with both positive and negative anomalies being associated with leptospirosis incidence higher than usual (Fig 6A). In opposition, precipitation showed a positive correlation for the case of New-Caledonia with lags of 0 and 1 month, and for Guadeloupe with a lag of 1 month (Fig 5). Precipitation appeared as a climate determinant of anomalies for New Caledonia with a best model including temperature (1 and 2 months lagged) and precipitation (1 month lagged) (Figs 6 and S3). Wetter climate led to an increased number of infections on this island (Fig 6B). However, we did not identify any precipitation variable driving inter-annual variations of leptospirosis in Guadeloupe. In this island, incidence anomalies were modelled solely based on temperatures with lags of 0 and 2 months (Figs 6 and S3). In Reunion Island and Guadeloupe, the incidence decreased with higher temperature while in New Caledonia, both negative and positive anomalies in temperature led to an increase in incidence (Fig 6). Mayotte was the only island for which precipitation with a 2-month lag was negatively correlated with incidence anomalies

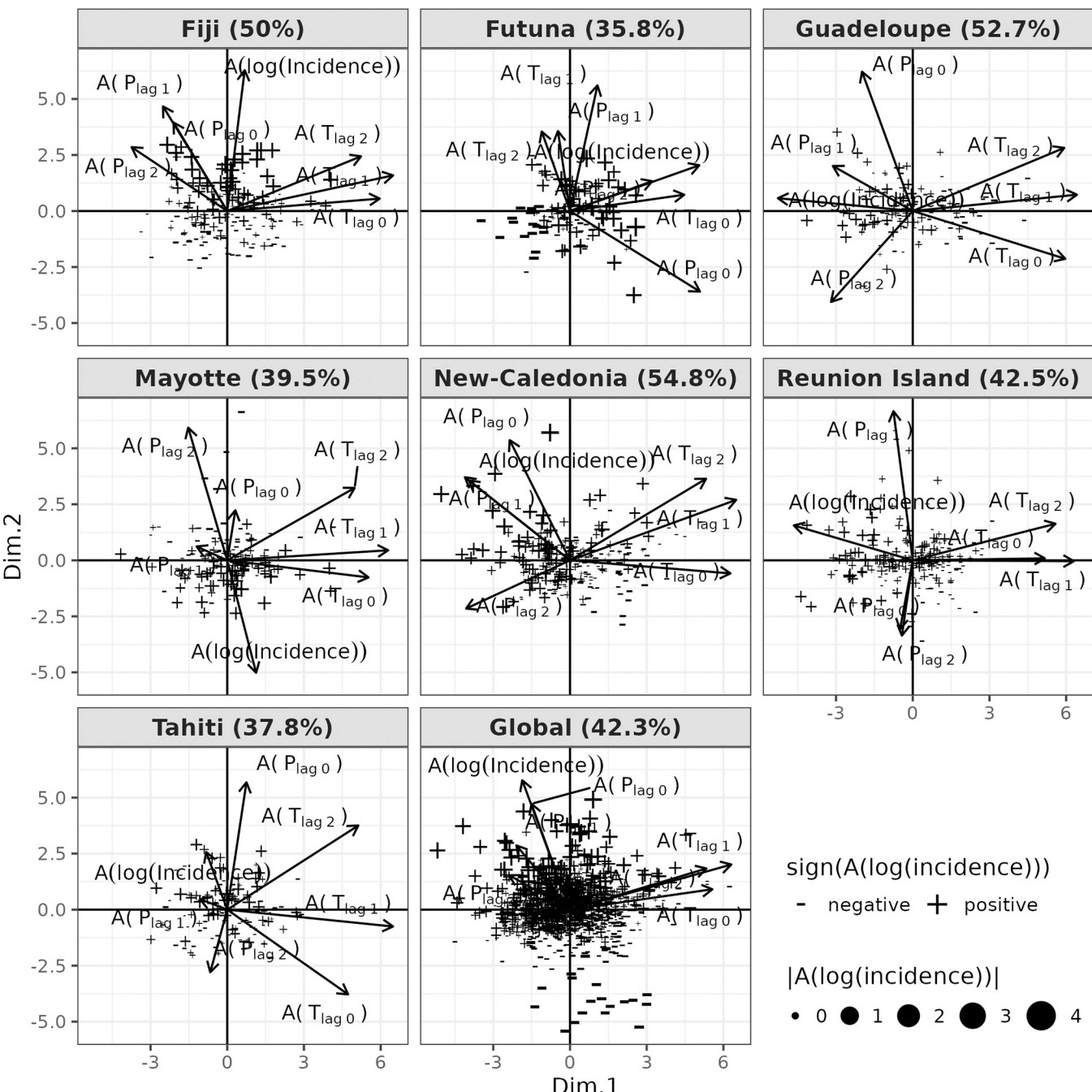

**Fig 5. Principal Component Analysis (PCA) for each island and the observed anomalies of log incidence.** The correlation of variables (arrows) and individuals are projected in the two first dimensions of the PCA. The percentage of variability explained by the two first dimensions of the PCA is given within brackets. Each individual of the PCA represents a month of the leptospirosis time-series. The sign (+/-) and the intensity (size) of the anomalies of log incidence were represented.

(Fig 5). The best model for leptospirosis in Mayotte indeed included precipitation with 0- and 2-month lag (Figs 6 and S3). In this island, both positive and negative precipitation anomalies decreased incidence (Fig 6).

The two first dimensions of the PCA explained a low percent of the data variability in Tahiti (37.8%), Futuna (35.8%) and Mayotte (39.5%) and poorly represented leptospirosis anomalies

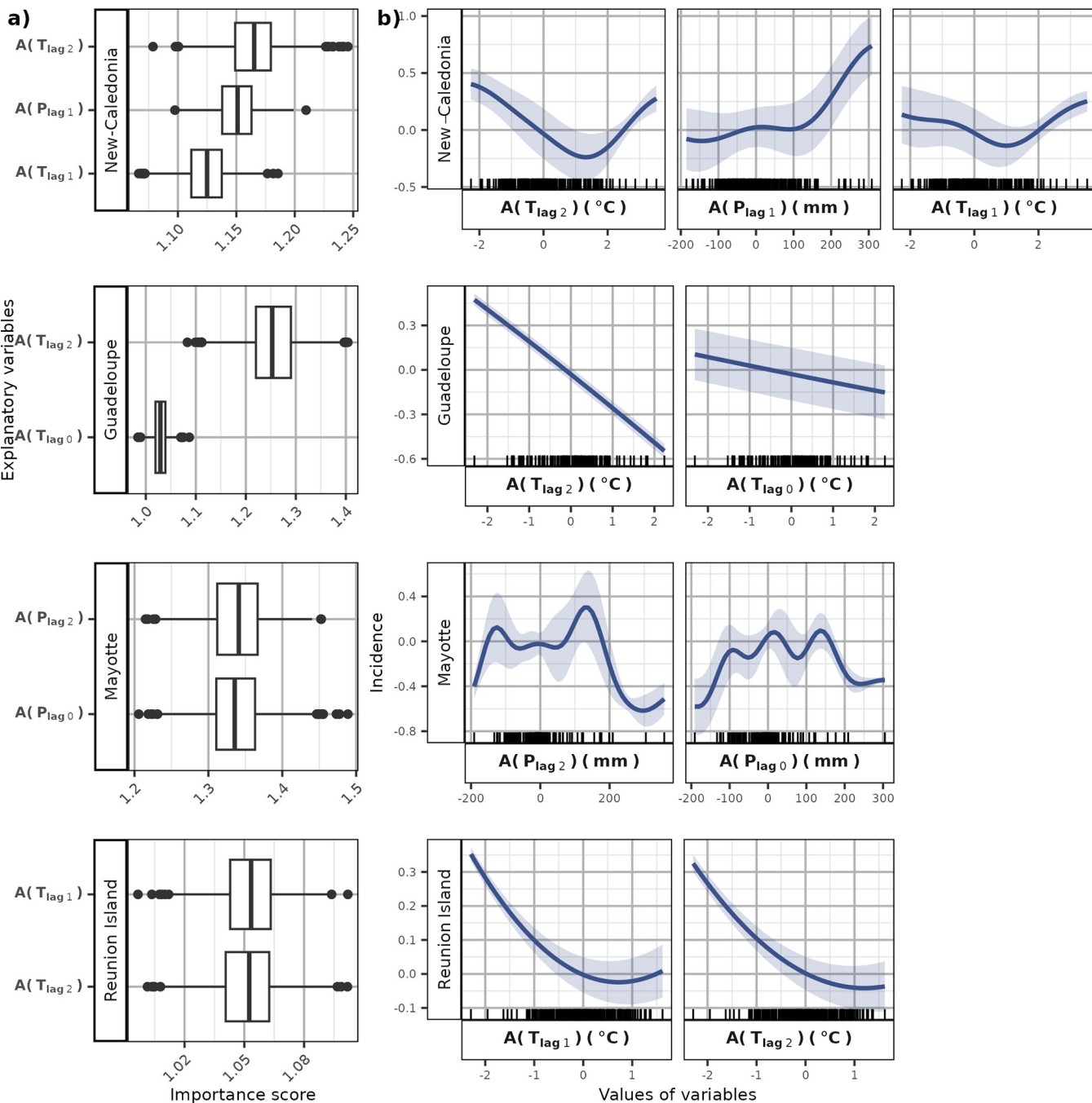

**Fig 6. Effect of the explanatory variable in the SVR inter-annual models adjusted on each island.** The importance score (a) represents the increase in MAE caused by repeated permutations of the variable in the dataset. The partial dependence curves (b) estimate the marginal effect of each variable by computing the average (blue line) (± standard error, blue ribbon) of ICEs curves (see methods).

for Tahiti and Futuna (Fig 5). The third dimension of the PCA revealed a negative correlation between anomalies and precipitations at lag 1 and 2 months for Tahiti and a positive correlation of the anomalies with temperature and precipitation at lag 0 in Futuna (S4 Fig). In Fiji, leptospirosis anomalies were poorly correlated with temperatures but showed great correlations with precipitation of the current month and at one month lag. We could not identify

climate determinants of anomalies in Tahiti, Futuna and Fiji since all tested models provided poor results in leave-year-out cross validation with low $R^2$ scores of respectively 0.21, 0.027 and 0.16 (Figs 2 and S3).

**Predictions of inter-annual local anomalies.** The inter-annual models, independently adjusted on each island, predicted well leptospirosis anomalies in Reunion Island, Guadeloupe, New Caledonia, and Mayotte. We computed the predictive abilities of models by performing leave-year-out cross-validation. The model of Guadeloupe showed the highest $\Delta MAE$ scores and improved the predictions of 14% compared to the seasonal profile alone (Fig 2A). In this island, the model correctly captured the 3 years of low incidence starting in 2014 and the higher incidence during the next 2 years (Fig 2A). In opposition, Reunion Island showed a low $\Delta MAE$ score of 5.8% and the model underestimated the outbreak in 2018 despite its good over-all adjustment ($R^2 = 0.47$ and MAE = 0.35) (Fig 2D). Although our model could accurately estimate both the low incidence between 2002 and 2006, and the two outbreaks occurring in 2008 and 2009 in New Caledonia ($\Delta MAE$ of 13%), outbreaks occurring in 2011 and 2021 were not predicted (Fig 2B). In Mayotte ($\Delta MAE$ of 13%), we overestimated leptospirosis incidence at the beginning of the time series, and we were not able to capture the peak in incidence in 2011 (Fig 2C). Despite these little differences with observed data, the models adjusted in Reunion Island, Guadeloupe, New Caledonia, and Mayotte globally improved the prediction solely driven by the seasonal profile ($\Delta MAE$ greater than 0) and could predict inter-annual variations occurring in these tropical islands.

## Discussion

In this study, we provided for the first time a global insight of the leptospirosis' dynamic over tropical islands. Our study encompassed Reunion Island, Mayotte, Guadeloupe, New Caledonia, Fiji, Futuna, and Tahiti into a global analysis to identify seasonal and inter-annual climate determinants of leptospirosis. We adjusted two types of models, a global seasonal model, and an inter-annual model specific to each island. These models helped identify common weather patterns leading to human disease cases in tropical islands and brought into light specific characteristics that impact the inter-annual variations of incidence. Moreover, our models provided the first estimates of leptospirosis' seasonal profiles in Vanuatu and Puerto Rico, where disease incidence data was not available, even though the disease has been recognized as a public health issue [21,22].

Time series of leptospirosis data brought into light a strong seasonality with higher incidence occurring between March and April for islands located in the southern hemisphere and between March and November for Guadeloupe. As commonly described in tropical areas, peak of incidence of this seasonal dynamic occurred during the hot rainy season [3,8–10,12]. The seasonal profile computed as the median incidence of each month and observed in Futuna did not draw a clear seasonal pattern. This island was therefore not included in the global seasonal model. A recent study conducted in Futuna drew the monthly cumulated number of cases of a 10-years period and revealed a seasonal pattern with highest number of infections occurring during the rainy season, the first half of the year [11], similarly to the dynamics observed in New Caledonia. We could not observe this pattern while using median profile because of the strong inter-annual variability of leptospirosis in this poorly populated island [11] that might have hidden any relationship with climate determinants.

As expected in a tropical setting [6], we identified rainfall as the main climate determinant of leptospirosis seasonality in the islands studied. In this model, rainfall of the current months and with a lag of 2 months was associated with an increase in human infections with a nonlinear relationship that was also evidenced in Salvador, Brazil [24]. Runoff caused by heavy

rainfall are prone to spread the bacteria throughout territories by washing contaminated soil and draining pathogenic leptospires into freshwater [4]. It can also favor the movement of reservoir animals increasing contacts with the human population [6]. Despite leptospirosis outbreaks have been monitored after extreme rainfall events and floodings [25], we did not identify rainfall as an important climate determinant of inter-annual anomalies in Reunion Island and in Guadeloupe. Large outbreaks usually occur during the rainy season. It was therefore possible that for these islands the environment was already at such a high risk for infections that more rainfall would not expand the contaminated environment. This effect was hypothesized in American Samoa, one of the wettest inhabited places of the world, where no association was found between local rainfall and leptospirosis seroprevalence [26]. In Mayotte, rainfall was expected to reduce incidence after a certain threshold. This effect was also observed in Thailand [27] and in Colombia [28] and suggests that extreme rainfall could also dilute and wash away the leptospires.

Temperature of the current and of the previous month also participated to model seasonal leptospirosis. In Reunion Island, the temperature of the current month has been shown to be positively correlated with the incidence [10]. However, while encompassing all islands into a single model to estimate seasonality, the marginal effect of temperature brought into light a negative relationship with the temperature of the current months. Surprisingly, temperature appeared as the most important climate determinant of leptospirosis inter-annual anomalies with a negative relationship with incidence in Reunion Island and Guadeloupe while both negative and positive anomalies increase leptospirosis incidence in New-Caledonia. As observed in Thailand, lower temperatures seemed more suitable for leptospirosis [27]. Optimal temperatures can favor leptospires growth [3]. Although desiccation induced by hot climate is not suitable for leptospires survival, it could promote water-based activity of both human and animal as bathing and drinking and intensify the sharing of persisting water sources increasing the probability of contamination [6,16].

Our models showed that climate lags ranging from 0 to 2 months prior to the onset of symptoms were important climate determinants of leptospirosis burden. These lags were consistent temporally with the leptospires survival into flooded lands and water-soaked soils [4] followed by the incubation period [3]. In our model, we focus on the short-term effect of climate events. Previous study identified ENSO-related long-lasting effects of climate that could impact leptospirosis inter-annually by building up the rodent reservoir resulting in an increased contamination of the environment and an increased seasonal rain intensity [17].

Although we achieved good predictions of incidence seasonal dynamics by building a global model of leptospirosis in tropical islands, we showed that 1) the incidence rate was difficult to capture in models and 2) climate determinants of inter-annual variability differed across islands, compromising the adjustment of a generic model. This suggested specific characteristics of islands cannot be neglected while studying leptospirosis dynamics. Our model did not include environmental indicators as water pH, soil type [4] and land use [6] that affect leptospires survival in the environment nor any consideration of the reservoir animals as rodents [29] and livestock [30] species both affecting the spatial distribution of the disease. Pathogenic leptospires have been detected in many mammalian species including domestic animals, livestock and wild animals [31]. In many islands the identification of the most common serotypes suggested that rats are the main reservoir of pathogenic leptospires [31]. In Reunion island, the dogs were also identified as a major carrier of the bacteria [10]. Futuna's economy relies on pig farming and taro cultivation and the agricultural practice increases human exposure to leptospires, with both rats and humans spreading the bacteria in the environment [11]. In Fiji the presence of rats, mice and mongoose at home did not appear as a significant risk factor while the presence of pigs in the community and cattle in the district significantly increased the odds

of being infected [30]. Livestock and domestic animals are restricted to specific areas (as farms, crops, backyard and private houses) and have an increased contact with the population as farmers, breeders, veterinary and pets sitters. In contrast, the spread of wild animals such as rodents is more difficult to control and contributes to the spread of bacteria in many different environments, from highly urbanized and populated areas to cultivated fields and very remote areas, potentially exposing a wide range of populations. Indication of the land use could therefore inform on the population exposure. The specificities of *Leptospira* animal reservoirs across islands is therefore likely to modify the local epidemiology of the disease temporally and spatially [31].

In addition, leptospirosis is recognized as a social disease whose occurrence depends on the behavior of the population, during professional or leisure activities, or in relation to cultural attitudes [3]. Our study area encompassed seven islands from different countries distributed around the world with different living conditions. In Mayotte, a large proportion of farmers allow their cattle to roam freely along dirt roads, fields, and rivers. In combination with poor living conditions and a lack of treated water at home, a quarter of the population turns to public fountains and rivers, which increases the risk of leptospirosis transmission [32]. Although no study has been conducted in these islands to link behavioral factors and infection, it is well established that impoverished rural-subsistence farmers and urban slum dwellers constitute the most vulnerable populations for leptospirosis [30]. Other risk factors recurrently reported include walking barefoot, swimming in streams and consuming water from different sources [16]. Agricultural activity either private or occupational appeared as the main risk factor of infection in French Polynesia, closely followed by the freshwater leisure activities [8]. Leptospirosis in Fiji was associated with outdoor occupations and poverty [30] whereas in New Caledonia most infections occurred during leisure or subsistence activities as hunting, fishing, bathing or swimming [33] and earlier peaks of incidence, caused by activities carried out during summer holidays, occurred in young age classes [34]. Leptospirosis is indeed increasingly associated with outdoor recreational exposure in developed countries. This change in exposure was expected to modify leptospirosis epidemiology in Reunion Island [10] and Guadeloupe [4]. The different lifestyles among tropical islands likely explained a part of the incidence not captured by our model solely based on climate indicators.

Our model accurately identified the high season of leptospirosis and the epidemic years. However, peaks of incidence remained underestimated. We used climate data issued from satellites images that, despite providing a reliable gridded estimation of climate over the territories, remain an indirect measure of the *in-situ* data (fragmented data). In addition, we spatially aggregated the climate data to match the resolution of the leptospirosis records available (monthly data per island) and estimate leptospirosis burden at the island scale. Highly localized and intense climate conditions not captured by our averaged proxy values could be related to an increase in infections. In Reunion Island, prevalence spatially varied within the island according to the annual rainfall level [10]. The use of *in situ* climate data could reveal particular conditions that increase incidence locally. In addition, as a recreational disease, epidemics can occur during punctual gathering events as the well documented outbreaks of the Lake Springfield Triathlon in 1998 [35] and more recently a triathlon in Reunion Island in 2013 [36]. Such events are likely not captured by general models and must be kept in mind while interpreting model's estimates, especially when working on small territories.

Moreover, the detection of leptospirosis cases relies on the surveillance system implemented which can strongly vary across countries and over time. Although our data solely included laboratory confirmed cases, leptospirosis cases must, first, be suspected clinically. Diagnosis therefore depends on the awareness of medical staff. The switch from a passive to an active surveillance of the disease multiplied the number of detected cases by 5 in Hawaii, when

testing patients presenting with at least 2 symptoms among fever, headache, myalgias, and nausea/vomiting [37] and by 8 in Futuna, when testing all acute fever [11]. It might partly explain the high incidence monitored in Futuna compared to the other islands. Increased communication and awareness might occur when an epidemic trend is discovered that leads to over-testing compared to the non-epidemics periods [8,25]. Symptoms of leptospirosis overlap with other tropical diseases, especially arboviral diseases, rendering diagnosis difficult. In 2018, a dengue outbreak occurred in Reunion Island [38] simultaneously with a peak in leptospirosis incidence. As observed in Hawaii during the period 2001–2002 [39] and in Puerto Rico in 1996 [40], dengue outbreaks might have contributed to increase detection of leptospirosis with dengue negative samples being tested for leptospirosis as differential diagnosis. In addition, periods with no or low incidence of dengue might promote leptospirosis as primary diagnosis and increase detection. Recorded incidence therefore strongly depends on the surveillance system implemented. Variations in case detection among islands and throughout time probably participated in the underestimation of outbreaks and impeded the adjustment of global models.

## Conclusion

Despite the complexity of the relationship between humans, environment, and reservoir animals, our results emphasize the importance of past and current climate factors, such as, temperature, and precipitation (lagged from 0 to 2 months), in the dynamics of leptospirosis. The key finding of this study is the identification of a unique climate-based model capable of describing the seasonal dynamics of leptospirosis in tropical islands across three oceanic basins on a global scale. This suggests that the seasonality of leptospirosis can be captured solely by climate indicators.

However, at the local scale, accurately estimating leptospirosis burden in tropical islands remains a significant challenge due to the diversity of lifestyles and of environments that influence the ability of climate to trigger outbreaks. While climate factors can easily estimate the seasonality of leptospirosis in tropical islands, a deeper understanding of local specificities, such as human behavior and environment characteristics should be investigated to improve our understanding of the occurrence of outbreaks inter-annually. The identification of climate indicators linked to leptospirosis dynamics represents a crucial initial step toward developing forecasting models. Future research should involve the investigation of a model at finer scale to capture the relationship between climate and disease dynamics at the local level and lay the foundations for the development of early warning systems for leptospirosis epidemics. Such models would inform decision-makers in the health sector and contribute to epidemic preparedness and management.

## Supporting information

**S1 Fig. Leptospirosis, precipitation and temperature seasonal profiles of the studied Islands.** The seasonal profile for Reunion Island (a), Guadeloupe (b), Tahiti (c), Fiji (d), New Caledonia (e), Mayotte (f) and Futuna (g) are defined by the mean temperature (*˚C*—red line) and the mean precipitation rate (*mm/hr*—grey bar) of each month (left panel). Leptospirosis profile is given in incidence per 100,000 inhabitants and was defined as the median value of incidence of each month (blue line).
(TIF)

**S2 Fig. Leave-island-out MAE scores for the tested global seasonal models.** Leptospirosis normalized seasonal profile was estimated based on the normalized seasonal profile of

temperature ($S(T)$) and precipitation ($S(P)$) with a lag ranging from 0 to 2 months. We performed normalization by removing the mean of the variables in each island. The black dotted boxes frame the best model selected for modelling the leptospirosis seasonal dynamics.
(TIF)

**S3 Fig. Leave-year-out $\Delta MAE$ scores of the per island inter-annual models.** Models estimate the anomalies of log incidence based on precipitation and temperature anomalies (respectively $A(P)$ and $A(T)$) with a lag ranging from 0 to 2 months. $\Delta MAE$ compares the accuracy of the predicted incidence to the seasonal profile. The black dotted boxes frame the best models selected for modelling the leptospirosis anomalies in each island.
(TIF)

**S4 Fig. Third dimension of the principal component analysis plotted against dimension 1 and 2 for Tahiti and Futuna.**
(TIF)

## Acknowledgments

We thank the various stakeholders that provided leptospirosis data: the Institut Pasteur and the department of Health and Social affairs of New Caledonia, the Regional Office of the French Institute for Public Health surveillance Antilles-Guyane (Cire), the French National Reference Center for leptospirosis, the Direction of Health of French Polynesia, Dr Mike Kama from the Fiji ministry of Health, Clément Couteaux from Agence de Santé de Wallis & Futuna.

## Author Contributions

**Conceptualization:** Léa Douchet, Christophe Menkes, Vincent Herbreteau, Morgan Mangeas.

**Data curation:** Christophe Menkes, Vincent Herbreteau, Cyrille Goarant.

**Formal analysis:** Léa Douchet, Joséphine Larrieu, Margot Bador.

**Investigation:** Léa Douchet, Christophe Menkes, Vincent Herbreteau, Cyrille Goarant, Morgan Mangeas.

**Methodology:** Léa Douchet, Christophe Menkes, Vincent Herbreteau, Cyrille Goarant, Morgan Mangeas.

**Supervision:** Cyrille Goarant, Morgan Mangeas.

**Validation:** Léa Douchet, Vincent Herbreteau, Cyrille Goarant.

**Visualization:** Léa Douchet, Joséphine Larrieu, Margot Bador.

**Writing – original draft:** Léa Douchet, Joséphine Larrieu.

**Writing – review & editing:** Léa Douchet, Christophe Menkes, Vincent Herbreteau, Joséphine Larrieu, Margot Bador, Cyrille Goarant, Morgan Mangeas.

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
