## [Decision Letter · Decision Letter 0]

13 Jan 2024

Dear Ms Douchet,

Thank you very much for submitting your manuscript "Climate-driven models of leptospirosis dynamics in tropical islands from three oceanic basins" for consideration at PLOS Neglected Tropical Diseases. As with all papers reviewed by the journal, your manuscript was reviewed by members of the editorial board and by several independent reviewers. The reviewers appreciated the attention to an important topic. Based on the reviews, we are likely to accept this manuscript for publication, providing that you modify the manuscript according to the review recommendations. 

Please read carefully the comments of the reviewers and address each one of them.

Sincerely,

Mabel Carabali, M.D., M.Sc., Ph.D.,

Academic Editor

Dileepa Ediriweera

Section Editor

Please read carefully the comments of the reviewers and address each one of them.

Reviewer's Responses to Questions

**Key Review Criteria Required for Acceptance?**

**Methods**

-Are the objectives of the study clearly articulated with a clear testable hypothesis stated?

-Is the study design appropriate to address the stated objectives?

-Is the population clearly described and appropriate for the hypothesis being tested?

-Is the sample size sufficient to ensure adequate power to address the hypothesis being tested?

-Were correct statistical analysis used to support conclusions?

-Are there concerns about ethical or regulatory requirements being met?

Reviewer #1: The selected methods are sufficient to support the results.

Reviewer #2: Table 1. Check the Monthly temperature of Guadeloupe: “3.226.8±1.4”, is a typing error? Also, subscripts are exchanged. “a” should be monthly precipitation and monthly temperature, whereas “b” should be the population description. 

Line 165: if median is reported and used for time series analysis, why the authors reported only the “mean” in table 1, maybe is better report both.

Line 173: link has an error. Please replaced.

**Results**

-Does the analysis presented match the analysis plan?

-Are the results clearly and completely presented?

-Are the figures (Tables, Images) of sufficient quality for clarity?

Reviewer #1: The results are impressive and are supported by the data.

Reviewer #2: Lines 231-232: why if Futuna is one of the islands with the highest incidence it is shown as SM and not in the main manuscript... If the manuscript is about 7 islands, why show the results of only 4?

Line 233: this should be supplementary material 3. Please reorder according as they appear in the manuscript.

Line 235: this is statistically different to the 2012 or 2014? the difference is 1 per 100,000 approx. as to say "the highest peaks".

Fig 3. I think it is important to include Futuna in this figure, even if the results are not as expected. Thus, in line 268-269 it is not necessary to refer to the supplementary material, but to the same figure discussed in the paragraph.

**Conclusions**

-Are the conclusions supported by the data presented?

-Are the limitations of analysis clearly described?

-Do the authors discuss how these data can be helpful to advance our understanding of the topic under study?

-Is public health relevance addressed?

Reviewer #1: The conclusions can be improved by pointing out the main findings of the present study and indicating points for future research.

Reviewer #2: Lines 418-423: discuss further the implications of these factors, as was done for social factors in lines 424-429.

Lines 435-437: the wording seems to refer to the same study, but two are cited (30 and 31).

I suggest reinforcing the conclusion, it is implied that the model is not useful. recommendations for the use of the model? who should use it? researchers, decision makers? health personnel? etc. 

Lines 450-451: any recommendations on the use of this model? The purpose of generating them is to prevent outbreaks? how should the social, geographic, climatic conditions mentioned be interpreted?

**Editorial and Data Presentation Modifications?**

Reviewer #1: (No Response)

Reviewer #2: Line 76: “from week to months”. From one week, or from weeks. Please edit.

Lines 89-90 how much is this influence? Please add some figures.

Line 109 poorly documented or notifiable. Please clarify.

Lines 110-111: add some reference to support this statement.

Line 274. please check and reorder all the supplementary material.

**Summary and General Comments**

Reviewer #1: (No Response)

Reviewer #2: (No Response)

PLOS authors have the option to publish the peer review history of their article (what does this mean?). If published, this will include your full peer review and any attached files.

Reviewer #1: Yes: Terencio Rebello de Aguiar Junior

Reviewer #2: No

Figure Files:

Data Requirements:

Reproducibility:

References

---

## [Editor Report · Decision Letter 1]

5 Apr 2024

Dear Ms Douchet,

We are pleased to inform you that your manuscript 'Climate-driven models of leptospirosis dynamics in tropical islands from three oceanic basins' has been provisionally accepted for publication in PLOS Neglected Tropical Diseases.

Best regards,

Dileepa Ediriweera

Section Editor

---

## [Editor Report · Acceptance letter]

18 Apr 2024

Dear Ms Douchet,

We are delighted to inform you that your manuscript, "Climate-driven models of leptospirosis dynamics in tropical islands from three oceanic basins," has been formally accepted for publication in PLOS Neglected Tropical Diseases.

Best regards,

Shaden Kamhawi

co-Editor-in-Chief

Paul Brindley

co-Editor-in-Chief
